



# Analysis of a risk prevention document using dependability techniques: a first step towards an effectiveness model

Laetitia Ferrer[1], Corinne Curt[1], Jean-Marc Tacnet[2]

[1]Irstea Aix en Provence, 13100, France
[2]Irstea Grenoble, 38402, France

*Correspondence to*: Laetitia Ferrer (laetitia.ferrer@irstea.fr)

**Abstract.** Major hazard prevention is a main challenge given than it's particularly based on information communicated to the general public. In France, preventive information is provided by the way of a regulatory
document named DICRIM (In French "Document d'Information Communal sur les Risques Majeurs" that means in English "Municipal Information Document on Major Risks"). It is made by mayors and addressed to the public in order to provide information on major hazards affecting their municipalities. Unfortunately, the law imposes only few specifications concerning its content therefore one can question the impact on the general population relative to the way it is concretely realized. Ergo, the purpose of our work is to propose an analytical
methodology to evaluate preventive risk communication effectiveness and apply it to the DICRIM. The methodology is based on dependability approaches. EFA (External Functional Analysis) permits the identification of (i) the service and technical functions involved, and (ii) the form, content and regulatory constraints of a DICRIM. FMEA (Failure Modes and Effects Analysis) is used to define the dysfunctions and detection elements are then listed to evaluate conformity with the 3 types of constraint. The outputs are validated
by experts from the different fields investigated. Those results are obtained in order to build in future works a decision support model for the municipality (or specialized consulting firms) in charge of drawing up documents. The method is applied to a database of 30 DICRIMs. This analysis leads to a discussion on points such as usefulness of the elements missing.

## 1 Introduction

Every year, major natural phenomena cause human and material disasters. Recently, in August 2016, an earthquake of magnitude 6.2 occurred in central Italy causing 250 deaths (*Huffingtonpost* 2016). Two months before, in the same year, the River Seine in France rose to a height of 6.10 meters and overflowed, causing 4 deaths, 24 injuries and a great amount of material damage throughout the different departments it traverses (BBCNews 2016). Preventive policies have been implemented to manage the consequences of these disasters,
such as the Hyogo or Sendai frameworks for action and disaster risk reduction (*UNISDR* 2015). Transmission of preventive information is equally important and has been the topic of current discussions highlighted by recent scientific researches (Newell and al. 2015) and international institutions as the United Nations (*UNISDR* 2015; United Nations 2006).

In France, numerous prevention systems and organizations exist to manage both natural and technological
hazards. Risk prevention requires the involvement of many stakeholders ranging from public authorities, experts and infrastructure managers to individuals and communities. In France, the legislation that relates policies and



prevention of natural hazards is imposed through six legislative texts. The first one, written in 1982, refers to compensating victims of natural disasters. Five year later in 1987, the need for emergency management is addressed and the right of access to preventive public information is recognized. In 1995, the creation of Risk Prevention Plans is required by law, which results in regulating urbanisation while taking into account major

hazards. In the following years, additional legislation continues to improve the management of major hazards (including technologic hazards). In 2004, the update and modernization of the 1987 law takes place. Notably, it insists on necessarily diffusing preventive information to the General Public (Observatoire Régionale des Risques Majeurs en Provence-Alpes-Côte d'Azur 2017). An example is demonstrated in Figure 1 which shows the regulatory mechanisms linked to urbanism and used for diffusing preventive information and its exchange

between different stakeholders. The thicker arrows indicate where the adopted requirements for increasing risk communication and risk awareness are applied. The different forms of communication (public meetings, information kits, posters, etc.) are used not only to provide safety recommendations but also as means of increasing individual knowledge of risk.

This is highly significant as human behavior during major disasters is influenced by their own knowledge of risk

(Enrico L. Quarantelli 2008). However, it equally depends on their cognitive bias (overconfidence, risk control delusion, denial, irrationality, etc.) (Kahneman and Tversky 1979) on the particular situation they face and more precisely on how they perceive that situation (Matt Dombroski 2006). Moreover, communication on risk can become a form of indirect experience of risk and thus a way of strengthening its acceptance and stimulating the involvement of exposed populations (Festinger 1957). If information is transmitted effectively people are more

prompted to adopt pertinent behaviour during the event as they have better knowledge of the associated risks and the safety recommendations for better risk prevention (Siegrist and Cvetkovich 2000). Results from a research study showed that information seeking by citizens seems to coincide with the intention to take preventive actions (Kievik and Gutteling 2011). Provide regulatory information to citizens can also lead to further information seeking (Hagemeier-Klose and Wagner 2009). Consequently the public needs to be periodically informed about

the hazards and the levels of risk they are exposed to and how their situation is changing (United Nations 2006).





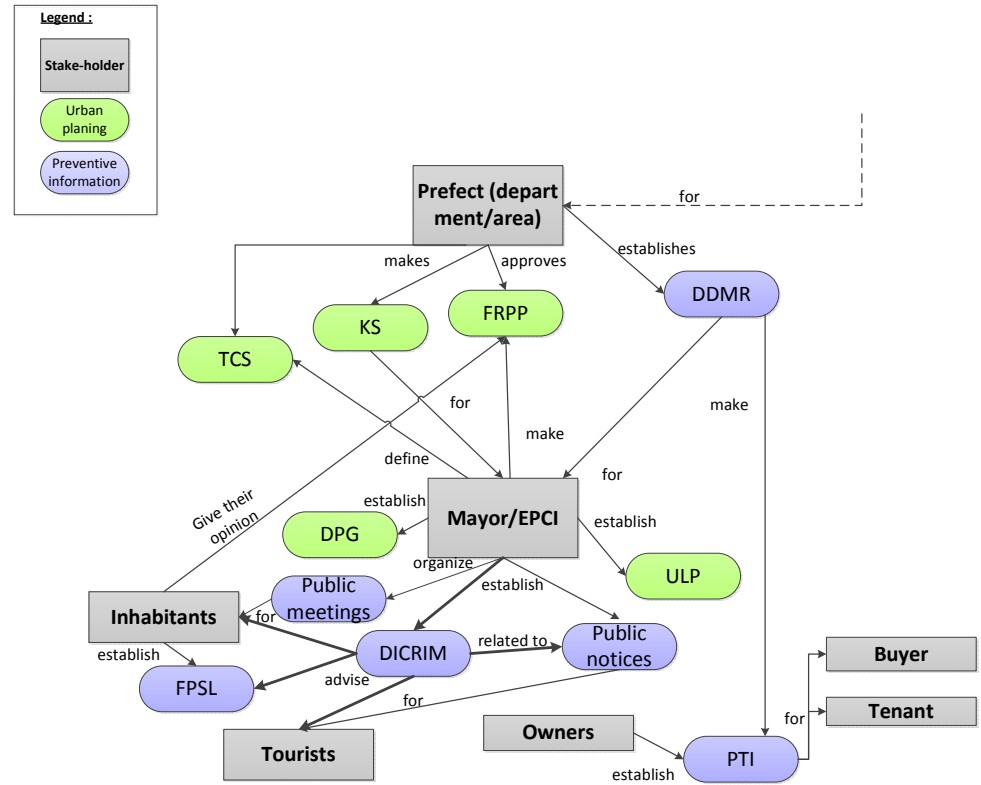

**Figure 1[1]: French regulatory prevention information throughout the town: Territorial Coherence Schemes (TCS); Knowledge Support (KS); Flood Risk Prevention Plan (FRPP); Development and Planning Guidelines (DPG); Urban Local Plan (ULP); Departmental Document on Major Risks (DDMR); Municipal Information Document on Major Risks (DICRIM); Public notices; Purchaser Tenant Information (PTI); Familial Plan for Safety Layout (FPSL); Public meetings**

Although significant progress has been made (recent decrees, investment by towns, more resources, etc.), certain features regarding risk information remain disappointing such as the uneven implementation and lack of control over these resources (IRMa and al. 2015), different behavior al instructions from one document to another even in reference to the same phenomenon (MAAF 2013), incomplete information about the hazards to which the population is exposed to and so on. It is also very difficult to establish if the system achieves its purpose in terms of being appropriated by the local population (risk culture) and especially if it achieves transformation (is behaviour changing?) (AFCPN 2013). Does it empower citizens with respect to knowledge on the risks that concern them? (Cutter 1993; Lindell and Perry 2004). To give such answers it is important to analyse the performance and effectiveness of such preventive communication tools while making sure that they reach their goal while conforming to the law.

---

[1] Prefect is a government representative of an area or department (France territorial division). It is thus responsible for public order, ensures the application of laws and regulations and verifies that the local authorities (Town Hall or EPCI which is a grouping of town halls) respect them as well.



In this work, we focus on preventive risk communication in regulatory documents (informal communication by social networks and by word of mouth is not considered) and more specifically on the DICRIM. Indeed, the councils of municipalities subjected to risks are obliged to provide preventive information to residents in the form of a DICRIM. Preventive information is also provided through other means but we chose to focus on
DICRIM because it is the main regulatory tool of compulsory form dedicated to the general public that summarizes all risks and their prevention. A brochure, film or advertisement may be just one part of a larger campaign to promote hazard readiness, and these items can and should be tested separately for efficacy and effectiveness prior to broad deployment (Sanquini, Thapaliya, and Wood 2016).

In general, inequality in terms of updating the specifications, transmitting the information and controlling the
execution of preventive information tools is observed and this concerns also the DICRIM. For example, in PACA region, in December 2016 only half of communities concerned have realized their DICRIM (ORRM 2016). Furthermore, the French government gives indications only on the general content of the document without details and without providing any standards. Works and writing guides have been proposed to develop DICRIMs (Clément and al. 2012) but the justifications of these recommendations are not clearly and fully
presented which leads to disparate documents in terms of content and form. The effectiveness of some DICRIM with the large number of pages with the citizens is also questionable, as is that of other DICRIMs containing a reduced number of pages, which can lead to oversimplification and significant loss of information. Finally, there is no link between quantity and compliance of DICRIM (Douvinet, 2013); having a DICRIM does not mean that municipalities provide good preventive information under the law (Rode, 2009).

Effectiveness of some contents in DICRIM is also questionable. For instance, the law required that DICRIM own maps regarding each hazard threatening the given town. Those maps are used to show to the population the hazard level geographically distributed on the territory of the municipality. But those maps are often similar to those elaborated in PPR, which is mainly used by experts, and they are not thinking to communicate to the Great Public. Flood maps are frequently seen as an information tool rather than a communication tool (Meyer and al.
2012).That is why some maps are really difficult to be understand by people which is regrettable because maps often firstly draws people attention before text. Some works proposes an evaluation of flood maps effectiveness notably based on interviews or eye-tracking and also give recommendations that should be used for hazard maps contained in DICRIM (Meyer and al. 2012; Fuchs and al. 2009).

These elements must be considered carefully because individual cognitive perception is reinforced or weakened by the form and contend of the provided informative document:; its communication is intended to be helpful and to result in the expected impact (Terpstra, Lindell, and Gutteling 2009). Consequently, assessing the effectiveness (form and content) of such legislative texts and the information they contain is an indispensable challenge, especially since this issue has received very little attention and only a few works on it have been
published to date (Gominet 2007; Kellens, Terpstra, and De Maeyer 2013).

For the purpose of rising to this challenge, this article has the objective of proposing an approach capable of analysing the DICRIM document, in terms of its effectiveness. The article is structured according to the following plan: it first examines selected methods for analyzing the effectiveness of communication tools; it then presents the methodology proposed after which the results of its application are shown, followed by an
application of part of these results to a DICRIM database. The article ends with the conclusion and perspectives.





**2 Analysis of candidate methodologies**

This issue requires the development and improvement of tools and methods for evaluating effectiveness. Such methods currently exist in different fields of study: advertising, geography, engineering, etc. They are generally

divided into three categories, according to whether: (i) they involve formal population samples; (ii) they do not involve formal population samples but rely on population perception assessment; (iii) they are based on systemic, analytical methods that do not involve the population.

Regarding communication and marketing, advertising is one of the major tools used by companies to

disseminate "persuasive" information on their markets, to inform, persuade and remind (McArthur and Griffin 1997). Methods have been developed to assess the effectiveness of advertising (pre- and post- advertising tests). These different methodological approaches based on questionnaires are of great interest. Current studies performed to evaluate the effectiveness of preventive risk communication (including DICRIM) for the public are classically conducted using the same methodological tools: surveys (either pre- or post-diffusion of the

document) by questionnaires. (Gominet 2010) and (Duaut and Luneau 2008), for example, obtained their results through prior surveys of mountain communities subject to natural hazards. In the same vein, the French Minister of Agriculture conducted a series of interviews with various stakeholders on subjects including risk, to measure the effectiveness of DICRIMs or to at least obtain their views on these documents (MAAF 2013). Moreover, communicating on risk can be considered as pertaining to "advertising" on how to protect oneself. As stated

above, the challenge is to make the public aware of risk, get them to accept it and protect themselves by taking appropriate actions (Renn 2014). The DICRIM must therefore have a persuasive effect on the population in the same way as advertising seeks to convince a consumer. If the questionnaires are well constructed and the results interpreted with rigor, these methods allow predicting the impact communication should have with regard to the pre-test and make it possible to modify the content prior to publication if necessary. In the post-test phase, they

also allow obtaining feedback on communication, on what worked or not, and on any dysfunctions in order to consider possible corrections and a new communication campaign. However, these methods also have limitations. Indeed, they require building a representative sample of the population that may require a budget and human and material resources which can be substantial. Moreover, these methods are focused on opinions and do not take into account factors that omit individuals' feelings.

Some methods do not require the use of a representative sample of the population but need indirect intervention:
- advertising efficiency standards for establishing performance comparisons, without it being necessary to use population samples;
- "connected" methods to evaluate the effectiveness of advertising such as the association of flyers, web sites and

/ or mobile applications (QR code, website audience, image recognition, etc.). For instance, a flyer can be effective if it has a QR code. It is then possible to quantify the number of "flashes" generated by people with their smartphones to visit the associated websites. The higher the number of visits, the more effective the flyer is deemed to be.
- ROI detection elements (return on investment) (click rates, visits, cookies, etc.) related to digital advertising

and tracking capabilities. They allow, for example, making post-tests a certain time after the broadcast of a message to identify individuals previously exposed to it.

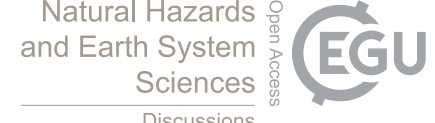

These evaluation methods appear relevant to our case study mainly because they do not require the intervention of inhabitants to assess the effectiveness of a DICRIM. However, there is an indirect need for some people to be involved by clicking or using QR code... If it can give an idea of the effectiveness about the retrieval of DICRIM it does not provide information about the impact of its reading. Is the information really received? Although

relevant regarding some aspects, these methods can be expensive.

Finally, the use of systemic, analytical methods based on structural, functional and dysfunctional analysis of the system under study seems relevant for the analysis. Their results could be used in the next works to build an evaluation model of effectiveness based on aggregated formalized indicators. Constructing an effective model will, of course, require involving the public but this won't be necessary when municipalities for instance arrive

to use the model. In such a sense, it could be financial gain for them. Effectiveness is the level of achievement of planned activities and achievement of expected results (Association Française de Normalisation 2005). It refers to the concepts of function and dysfunction. Thus analytical methods, such as dependability analysis used in engineering, seem relevant. DICRIMs are subject to complex processes involving interdisciplinary concepts (information processing, cognition, communication, etc.). The lack of a formal, detailed framework (form and

content), and lack of familiarity by the public make their effectiveness difficult to assess without a systemic and rational approach. Dependability methods allow the identification of risks and analysis of behavior and failures. These methods are qualitative (Preliminary Analysis of Hazards, FMEA, Summarized Breakdowns Combinations Method) and quantitative (Fault Tree Method, Event Tree Method, etc.), based on the construction of state graphs (Space States Method, stochastic Petri nets), and on simulation (Monte-Carlo simulation). They

were developed for complex industrial systems and applied in different domains (Peyras, Royet, and Boissier 2006; Zouakia, Bouami, and Tkiouat 1999; Bambara and al. 2015). These methods are interesting as they enable identifying the elements involved in the effectiveness of the system under study, determining the causes and effects of dysfunctions and listing detection elements able to pinpoint the occurrence of dysfunctions. Our work takes this direction and aims it at applying a function and dysfunction analysis method to a DICRIM. The

problem to overcome is that the methods chosen must be adapted to our context, which is not one of engineering, but one of decision-making. The first successful application of dependability methods in such a context was carried out in a field that interests us: the diagnosis of contingency plan failures (Girard and al. 2014; Piatyszek and Karagiannis 2012; Bambara and al. 2015). In other words, the systemic analysis of the DICRIM can provide the means of performing a qualitative analysis of effectiveness.

**3 Material and Methods**

**3.1 The system studied: the DICRIM**

The system studied is the DICRIM. It has been chosen because, in the French government's overall strategy for risk prevention, it is the main reference document in terms of informing the public about the natural and technological risks affecting the municipal territory. It is prepared by the mayor on the basis of the DDMR

provided by the Prefect (it lists all essential information on major natural and technological risks at the level of their department) (Figure 1). The municipalities required to produce a DICRIM are those in which there are Risk Prevention Plans and / or those designated by prefectural decree because of their exposure to a particular major risk.

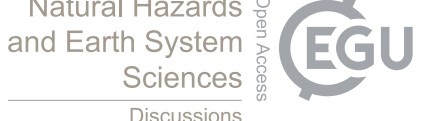



The main headings and information to be included in the DICRIM are listed in the National Model for the application of the Environment Code (Articles L 125 - 2 and R 125 - 9 to R 125 - 27) issued by the MTES (French Ministry of Ecological Transition and Solidary)(MEDDE 2013). Our analysis will be based on it. It is a framework with general information that must be used both by mayor and prefect to create respectively DICRIM

and DDRM. To do so they have to use this framework and to add specific information, for instance information regarding the municipal territory for mayor. Following the recommendations described in (31), the DICRIM is composed of: a word from the mayor, a DICRIM presentation, several information for each hazard affecting the town (risk presentation, prevention and protection actions, safety instructions, mapping), information about other preventive information devices (communal poster, flood marks…), emergency phone numbers and equipments

to always have at home to be ready (cf. Table 3). These are viewed as components in the systemic analysis. The major risks that must be dealt with in the DICRIM (according to their occurrence in the given town) are listed in the National Model and are as follows: floods, earthquakes, ground movements, forest fires, avalanches, storms/cyclones, volcanic eruptions, dam failures, nuclear accidents, industrial accidents and mining risks. There is no legal obligation regarding the document's format. The DICRIM can be consulted by the public at the

town hall as a paper document. In some towns it is also distributed directly in the mailboxes of inhabitant or posted on the municipal website in digitized form or more rarely, in interactive mode.

**3.2 Global approach**

The method proposed is an analysis of the system that considers its functioning and dysfunctioning (Cf. Figure 2). It consists of a functional analysis that fully describes the functions and relationships in the system and the

constraints it fulfills. A structural analysis allows determining two levels: the DICRIM (system itself) and the components listed above.  Functional analysis is performed at these two levels: external concerning the global system and internal carried out component scale. Functions are systematically characterized, classified and evaluated (AFNOR 2014). For the functional analysis we chose to use the APTE method that can be applied successfully to our case study as a quick test, as proposed in (Ghariani, Curt, and Tacnet 2014). It is one of the

dependability methods used most and generally provides the basis for a subsequent Failure Mode Effects Analysis (FMEA) which is the last part of the approach.

FMEA is an inductive method of analyzing potential failures in a system. It systematically considers each system component and its failure modes one after another. Failures are identified by non-compliance with constraints.

Detection elements are also listed, allowing the identification of this non-compliance.  The results of FMEA analyses are presented in tabular form, specifically designed for the type of system studied. The table columns are: "components", "technical function", "failure mode", "possible cause of failure", "detection element" and "possible effects of the failure". The detection elements were formulated by making use of the literature. For example, the field of advertising was investigated for the part concerning the DICRIM's form.


A database was built and analyzed to enrich the FMEA results. The latter were also validated by experts. When performing an FMEA in an industrial situation, classically specialists in different areas are involved in completing and validating the results obtained. Several domains are necessary to evaluate the effectiveness of preventive from risk and communication: that justifies the choice we made for experts in our work. Three

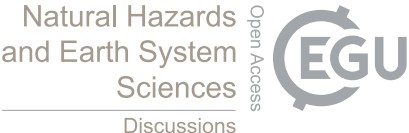

experts were asked to carry out this task: two are experts who have performed research in technological or natural risk analysis for more than 15 years, while the third is a communication expert with 27 years of experience in the field.

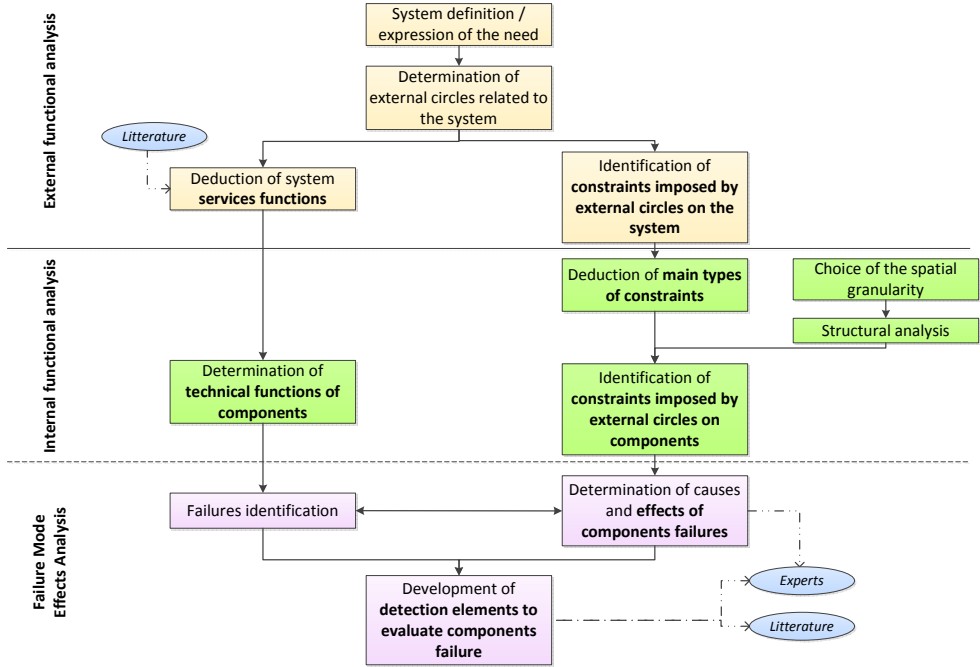

**Figure 2 : Detailed steps of Functional Analysis and FMEA**

The main outputs of the approach are functions (technical and service), constraints and the main types of constraints, failure effects and detection elements. They are shown in bold in Figure 2.

All those steps are executed in order to analyze the performance of the DICRIM which is considered part of its

effectiveness. Simultaneously, another evaluation is conduced. During the study of the system needed to carry the methodology presented before, an analysis of regulatory requirements was carried out. Indeed, it was mentioned earlier that government provides some general instructions regarding the content of DICRIM in the form of a national model. We can think that this model has been strongly discussed in order to make the DICRIM as effective as possible. That is why in order to analyze its effectiveness we must also analyze the

DICRIMs compliance to the law by identifying detection elements once more.

Later, detection elements will be formalized as indicators. An indicator is information that helps a stakeholder, an individual or a community in general, to carry out the course of actions needed to achieve a goal or to evaluate a result (Bonnefous and Courtois 2001). It should be formalized in order to make its use repeatable and reproducible (Curt, Peyras, and Boissier 2010). In future work, a more detailed analysis will be performed to

20 combine certain detection elements. They will be aggregated with each other to form a model that will give an effectiveness score as output. Feedbacks will also be provided by the model to know which component must be improved and how. Figure 3 shows the global approach of the methodology we propose.



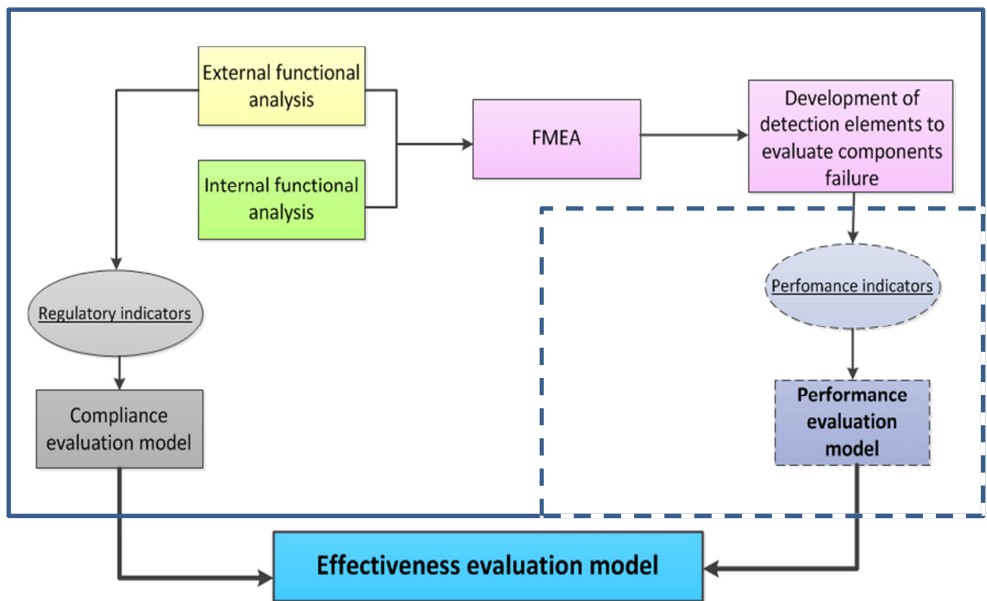

**Figure 3 : Global methodology proposed and followed to build the effectiveness evaluation model**

The results shown below stem from the application of the method to the case of the DICRIM the process of which is presented in Figure 3. Performance indicators and the construction of a performance evaluation model as shown with a dashed line represent our future work for which only some examples will be shown in this paper.

## 4 Results

### 4.1 External functional analysis

The EFA consists of 3 main steps: system definition, expression of needs, and determination of relationships with external environments.

#### 4.1.1 System definition and needs satisfied by the system

First, it is necessary to define the system to be studied precisely. Limits are defined, which then lead to considering the interactions with external environments. The system therefore takes into account the document (DICRIM) with its content, form and accessibility as described by the relevant law.

EFA is used for translating the need satisfied by the system.

In general, it can be formalized by three questions applied here to the DICRIM: (1) To whom is the system dedicated? - Response: To the General Public; (2) On what does it act? - Response: On its knowledge and risk perception; (3) Why is this action necessary? (that is to say, for what purpose does the system exist?) - Response: To inform on hazards and on how to act when phenomena occur.

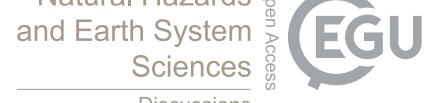

**4.1.2 External circles**

It is then necessary to determine the external circles that interact with the system. They are made up of human elements, natural elements or part of other systems which can act on the document or be subject to its actions. The inventory of these external circles (designer, director, regulations, residential accommodation, etc.) is
established by examining the environment of the DICRIM. We also highlight interactions between these areas and the DICRIM. Figure 4 illustrates the design, construction and operation process (reading and use by the public). Interactions are materialized between the system and its environment using a functional block diagram, in which we distinguish the contact relationships (represented by straight segments) and flows (represented by arcs).    Interactions between an external circle and DICRIM identify constraints to be considered. Flow
relationships between 2 external circles identify the service functions performed by the system. The quality of the external circles and the analysis of interactions determine the completeness of the functions obtained.

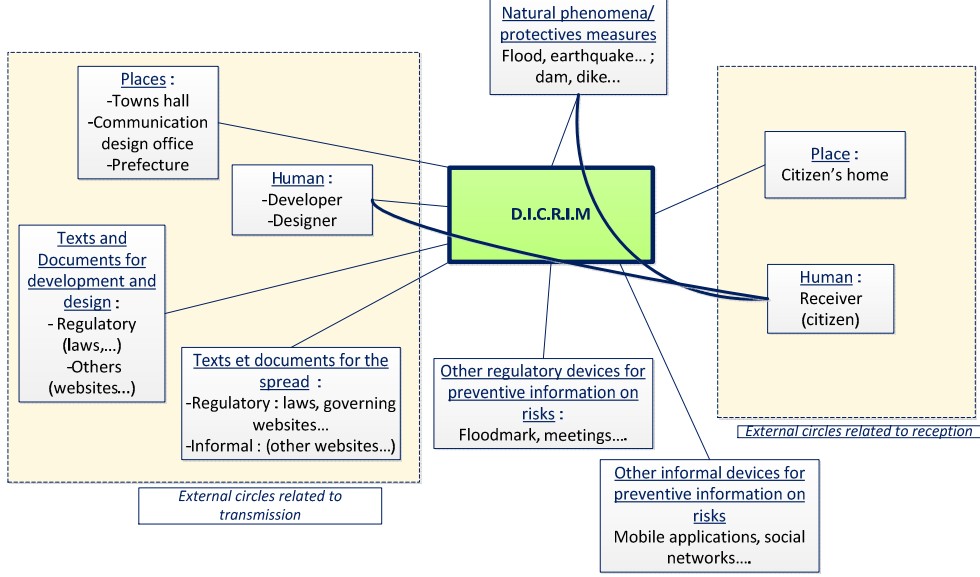

**Figure 4 : Functional block diagram for DICRIM**

**4.1.3 Service function and constraints**

Service functions reflect the objective of a system. They are determined through relations between two external environments. Figure 4 shows that the transmitter (developer and designer) is linked with the receiver through the document. This is the process involved in preventive information on hazards. With the DICRIM, the receiver, that is to say the public, is also related to major events when they happen. They must implement what
they learn from the document in terms of behavior. Two DICRIM Service functions are then deduced:

1) SF1: the DICRIM informs on maintaining and fueling risk culture (learning in the operational phase). We consider that risk culture is fueled by 4 main elements: risk awareness, knowledge, acceptance and memory (Johnson 1993; Terpstra 2011).





2) SF2: the DICRIM informs on acting appropriately when facing a major event (decision-making in the operational phase). An appropriate act is based on the way the event is detected and good practices (Bostrom and Löfstedt 2003).

These are functions fulfilled by the document to satisfy the constraints exerted on it by external circles. A

5 constraint is a "characteristic, effect or design provision that has been made mandatory or prohibited for any reason whatsoever. No other possibility is left" (AFNOR 2015). Constraints are obtained by examining the external circles in interaction with the document. They are listed in Table I (extract).

| Types | External cycles | Constraints (DICRIM must...) |
|---|---|---|
| *Places* | Inhabitants home | -have been received<br>-have been keep |
| | Town hall | -to be designed<br>-to be available<br>-to be diffused |
| | Communication design office | -to be developed |
| | Prefecture | -to be archived |
| *Human* | Designer | -to be correctly designed in order to respect Human/Designer; Human/Receptor; Natural phenomena/protectives measures ; Texts and Documents for development and design constraints |
| | Developer | -to be correctly developed in order to be correctly designed in order to respect Human/Designer; Human/Receptor; Natural phenomena/protectives measures ; Texts and Documents for development and design constraints |
| | Transmitter | -widely spread<br>-respect «have been received» by Places/Inhabitant's home |
| | Receptor | -To be designed and developed to capture attention<br>-To be designed and developed in order to persuade receptor to read the DICRIM<br>-To be designed and developed in order to promote understanding and recording of information |
| *Major phenomena/protectives measures* | -Floods, earthquakes, ...<br>-Dams, dikes... | -Inform on major phenomena and on protection measures taken by town to overcome them. |
| *Texts and Documents for development and design* | Laws | -Respect laws |
| | Model | -Follow national model given by MTES |
| | Governmental Websites | -Respect safety instructions or recommendations they contain |
| | Departmental Document on Major Risks (DDMR) | -Synthetize and adapt information contained in the DDMR to general public |
| *...* | | .... |

10 **Table I : Constraints features highlighted by the interactions between the DICRIM and external circles (extract)**





As shown by Table I, constraints can be classified into 4 main groups: regulatory obligations ("compliance with laws", "in line with the national model given by MTES"…), substance constraints ("inform on major phenomena…"), form constraints ("to be correctly designed"…), and information flow ("have been received", "have been kept"…). Three of these groups, regulatory/substance/form, will be kept to analyze the constraints at

5 the component level (internal functional analysis). At this scale, we will not consider the fourth group (information flow) because it does not concern the component level. It is taken into account at a higher level of granularity when the global system is considered.

## 4.2 Internal functional analysis

Technical functional analysis (or internal) (TFA) is the part of functional analysis that helps to formalize and study the architecture of the product (structural analysis) and identify the technical functions of the components (AFNOR 2015).

### 4.2.1 Components / Service functions

Structural analysis aims at listing all the components of the system. The DICRIM components are identified in

the national model issued by the MTES. There are at least 16 components (cf. Table 2). More components are possible because 6 among those 16 (Components n°5 to n°10) are applied for each hazard encountered in the town studied; the 10 elements out of 16 remaining do not depend on hazard. For instance, if two hazards threaten the town, there will be 22 components (10 + 2x6). That is why there can be as many as 76 (10 + 11x6) components in the DICRIM if a town faces all the hazards listed in the national model (11 hazards, consult §

3.1).

Municipalities are free to introduce non-legislated components; however, we do not consider this option at this stage of the analysis. These regulatory components are listed in Table II according to whether they concern the first service function or the second.

| Component | Service function 1 (SF1):<br>*Informs to maintain and fuel risk culture.* | Service function 2 (SF2):<br>*informs on acting appropriately when facing a major phenomenon* |
| --- | --- | --- |
| **Cover page (Cp1)** | x | |
| **Editorial with a word from the mayor (Cp2)** | x | |
| **Summary (Cp3)** | x | |
| **DICRIM presentation (Cp4)** | x | |
| **Presentation of the risk in the town (Cp5)** | x | |
| **Prevention actions at town level (Cp6)** | x | x |
| **Police and protection actions** | x | x |





| | | |
|---|---|---|
| **(Cp7)** | | |
| **General safety instructions (Cp8)** | | x |
| **Specific safety instructions (Cp9)** | | x |
| **Mapping 1/25.000th (Cp10)** | x | |
| **Communal poster (Cp11)** | | x |
| **Flood marks and highest known flood zone (Cp12)** | x | |
| **Underground cavities and marl pits(Cp13)** | x | |
| **Where to get more information (Cp14)** | x | |
| **Emergency phone numbers (Cp15)** | | x |
| **Equipment to always have at home to be ready (Cp16)** | | x |

**Table II: Components and links with service functions**

Some components are specific to one or the other service function. For instance, components 1 to 5 are specific to SF1. On the contrary, components 15 and 16 are specific to SF2. Indeed, they correspond to emergency phone numbers and equipment, two elements used in case of an event to act appropriately. Some components also

perform both service functions. This is the case, for instance, of "policy and protection actions" (Cp7) because they participate in fostering the acceptance of risk, which is one of the elements of risk culture characterizing SF1. Indeed, protection actions result in explanations of departmental rescue plans, for example, which make people realize that the risk must be taken seriously. CP7 also performs SF2 insofar as it also gives information on the way the public will be alerted if an adverse event occurs. Alerting is one of the factors that characterize

SF2. Finally, all the components are linked to at least one service function.

### 4.2.2 Service Functions / Technical Features

Each component has one or more functions, so-called technical functions. They contribute to the service functions. Moreover, they must satisfy the constraints. The technical functions are now linked to the different

components: a component fulfills one or more technical functions (Figure 5).




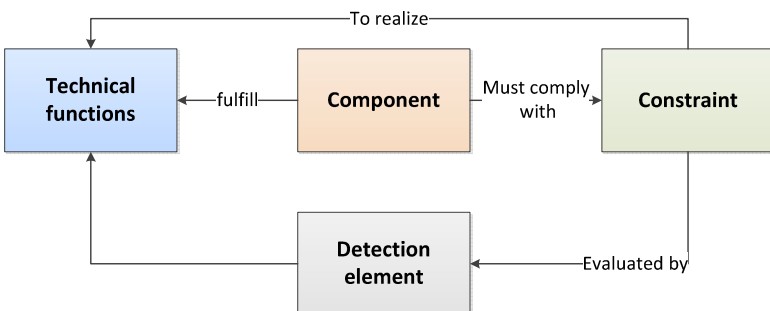

**Figure 5 : Role of the entities of the method.**

When establishing the two service functions of the DICRIM, some elements were identified as being linked to
5  both of them. Here, these 6 identified elements are expressed in terms of 6 technical functions: 4 contribute to
risk culture and 2 concern actions during an event. The technical functions of all the components are listed in
Table III according to whether they concern the first service function or the second.

| Service Functions | Technical Functions |
|---|---|
| *Informs to maintain and fuel risk culture* | -Inform to raise awareness of risk (TF1) -<br><br>-Inform to fuel knowledge on risk (TF2)<br><br>-Inform to foster the acceptance of risk (TF3)<br><br>-Inform to maintain risk memory (TF4) |
| *Informs as to the appropriate action to be taken when facing a major phenomenon* | -Inform on event detection (TF5)<br><br>-Make known the appropriate reflexes to adopt if a phenomenon occurs (TF6) |

**Table III: list of technical functions identified for both service functions of DICRIM**

The identification of the technical functions is based on the following elements. The "culture of risk" is not to be
seen as a more or less distributed capital but rather as a pragmatic relationship to the danger that is constructed




and rebuilt perpetually, sometimes individually, sometimes collectively (CEPRI 2013). It begins by risk awareness. Knowing the flood does not imply feeling directly concerned by this risk. This awareness of risk has a subjective dimension, specific to each individual or group (leads to define TF1). Conversely, the awareness of what constitutes risk to oneself or to a group can not be effective without some knowledge of this risk. This

knowledge, theoretical and practical and corresponding to TF2, is also built up over time, in particular by means of information received formally or informally, such as the reception of a DICRIM for example. Knowledge and awareness about risk need to be maintained in time, so that it can be forgotten. For a variety of reasons, the transmission of ancestral knowledge has gradually been extinguished. New populations from urban areas and tourists are often unaware of the risk their municipality is exposed to. Moreover, a phenomenon may possess a

very long period of return, even leading the long-settled populations to the same place forgot the danger still present. It leads to the TF3 characterizing memory of risk. To hope an appropriated behaviour, acceptance of risk is also crucial (TF4). Again, this process is strongly related to the nature of the risk communication that has been carried out, including the credibility of the source. When an individual feels vulnerable to risk, feeling that he or she is deprived and without means to cope with risk, he or she has more difficulty accepting the risk and tends to

take refuge in the denial of that risk. Denial is part of a set of perception biases (overconfidence, anchoring effect…) which can occur in the face of risk, which runs counter to the ingredients described above and which build the culture of risk (awareness, knowledge, memory, acceptance) (Figure 6).

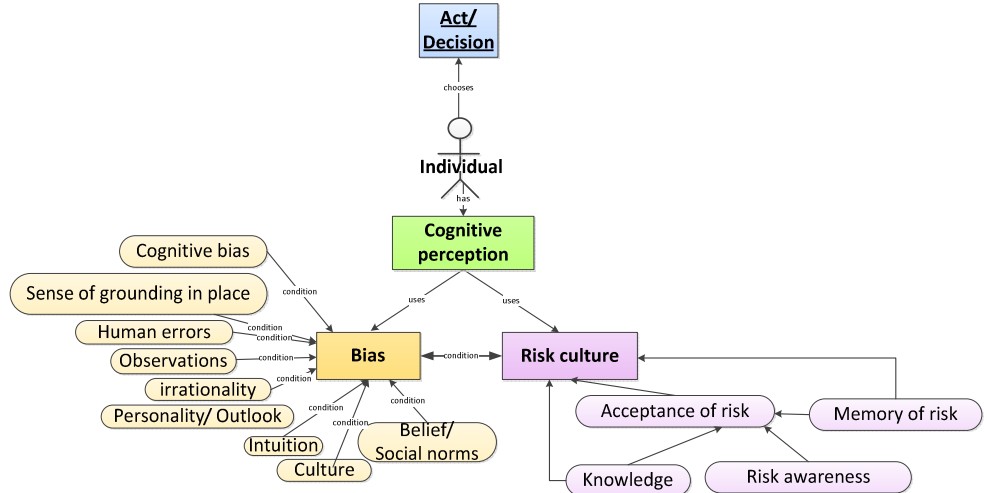

**Figure 6 : Factors influencing a decision-making process related to the occurrence of a phenomenon**

In many cases, cognitive bias takes precedence over the rationality of a decision. For example, over-confidence corresponds to an overestimation of his knowledge or an underestimation of his uncertainty. Most of the time subjects underestimate what they do not know. Preventive risk information systems such as the DICRIM therefore exist with a view to avoiding as much as possible a bias take precedence in decision-making context.

The DICRIM would therefore contribute to the risk culture by deepening these major ingredients by using technical functions. The DICRIM also informs on acting appropriately when facing a major event. An





appropriate act is based on the way the event is detected (leading to TF5) and good practices characterizing TF6 (Bostrom and Löfstedt 2003).

### 4.2.3 Technical features / Components

5 Table IV represents the membership of each technical function identified for the 16 DICRIM components. We consider that components 5 to 10 generally concern each hazard. Generally, all the functions are covered by at least 5 components and all the components perform at least one function. TF1 is filled with ten components. This is the function performed by most of the components. Cp14 satisfies all the technical functions of both service functions. Components 8, 9, 15 and 16 perform only one function (TF6). The pairs of components 1/2, 3/4 and 10 12/13 fulfill the same functions. Cp5 performs all the technical functions linked to SF 1 and CP 11 the two technical functions of SF 2.

| | Cp 1 | Cp 2 | Cp 3 | Cp 4 | Cp 5 | Cp 6 | Cp 7 | Cp 8 | Cp 9 | Cp 10 | Cp 11 | Cp 12 | Cp 13 | Cp 14 | Cp 15 | Cp 16 |
|---|---|---|---|---|---|---|---|---|---|---|---|---|---|---|---|---|
| TF 1 | x | x | x | x | x | x | | | | x | | x | x | x | | |
| TF 2 | x | x | x | x | x | x | x | | | x | | | | x | | |
| TF 3 | | | x | x | x | | | | | x | | x | x | x | | |
| TF 4 | | | | | x | x | | | | | | x | x | x | | |
| TF 5 | | | | | | x | x | | | | x | | | x | | |
| TF 6 | | | | | | | | x | x | | x | | | x | x | x |

**Table IV: membership of each technical function identified for the 16 components**

The redundancy in this table is obvious. However, hasty conclusions must not be drawn from these observations. For example, although two components fulfill a single function, this does not mean that one duplicates another.



Depending on the context, fueling the same function by several components may be beneficial. For example, awareness may not be immediate and require the addition of more information. Conversely, too much knowledge can cause boredom for the reader who may lose interest in the topic or the whole component. A detailed analysis will be performed when identifying adjustments to the content of the regulatory document. This concerns works

in progress, not presented here, that consists in interviewing the population about their perception of the DICRIM.

### 4.2.4 Constraints / components

To fulfil its technical function, a component must conform to one or more constraints. The respect for the

constraints by components are evaluated by detection elements that will be showed later in the article. The first constraints are regulatory ones. In the case of the DICRIM, regulatory constraints are highlighted in the national model issued by the MTES. But during our analysis we found that these constraints were not sufficient when the aim is to evaluate the effectiveness of the DICRIM. Three groups of constraints were identified in the EFA when listing the constraints exerted on the DICRIM by external circles. The same groups have to be considered here

regarding the scale of the components. That is why we have added new types of constraints in addition to the regulatory ones. They are substance constraints and form constraints identified in the literature (communication, advertising, etc.). It was necessary to add form constraints, in particular, because the national model does not include any specific instruction on this part whereas it plays a crucial role in the effectiveness of a communication. Table V shows an extract of constraints filled by each component relative to substance

characteristics. Table V is composed of a total of 9 lines. Each component satisfies at least one constraint. The first three DICRIM components must persuade the reader to continue reading the DICRIM because they are usually the first items they see when they open the document. There are, however, no regulations on these three components. The designer is therefore free to develop the DICRIM. Each component must capture attention in order to be read.

The same type of table was produced for regulatory constraints (13 lines) and for form constraints (6 lines).

| Substance constraints | Cp 1 | Cp 2 | Cp 3 | Cp 4 | Cp 5 | Cp 6 | Cp 7 | Cp 8 | Cp 9 | Cp 10 | Cp 11 | Cp 12 | Cp 13 | Cp 14 | Cp 15 | Cp 16 |
|---|---|---|---|---|---|---|---|---|---|---|---|---|---|---|---|---|
| **Persuade to read the DICRIM** | x | x | x | | | | | | | | | | | | | |
| **Capture attention** | x | x | x | x | x | x | x | x | x | x | x | x | x | x | x | x |



| Promote understanding and recording information to maintain and fuel risk culture | x | x | x | x | x | x | x | | x | | x | x | x | | |
|---|---|---|---|---|---|---|---|---|---|---|---|---|---|---|---|
| Promote understanding and recording information to act appropriately when facing a major phenomenon | | | | x | x | x | x | | x | | | x | x | x | |
| … | | | | | | | | | | | | | | | |

Table V: Substance constraints satisfied by each component (extract)

### 4.2.5 Functional analysis table

For each component, the technical functions along with the service function they refer to and the constraints are then gathered in Functional Analysis Tables. Table VI presented below for the component "Editorial with a word from the mayor" is an example:

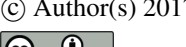



| Component 2 | Service function | Technical function | Constraints |
|---|---|---|---|
| **Editorial with a word from the mayor** | *Informs to maintain and nurture the culture of risk* | -Inform to raise awareness of risk<br><br>-Inform to foster the acceptance of risk | Form : -To be designed and developed to capture attention<br>-To be designed and developed in order to persuade the receptor to read the DICRIM |
| | | | Content : -To be designed and developed in order to persuade the receptor to read the DICRIM<br>-To be designed and developed in order to reassure the reader<br>-Introduce DICRIM function<br>-Explain importance of risk management |

**Table VI : Functional analysis table for Component 2**

**4.3 Failure Modes and Effects Analysis table**

The results produced by the functional analysis form the basis for performing an FMEA. The FMEA is carried out for each function performed by the components of the document. All the failure modes (that is to say

5  functions not carried out) that can occur during the different life cycles of the document and their causes and effects are identified. It is essential to investigate the causes and effects of malfunctions carefully to detect them and then propose feedback to avoid them. Effects are not used in our work but they will be important results in the future steps of building the model of evaluation effectiveness. Furthermore, as we said before, a component must satisfy a constraint to perform a technical function. Non-compliance with the constraints is seen here as a

10  cause of dysfunction (cf. Figure 5) and they are therefore essential to identify detection elements. Compliance with the constraints is evaluated by one or more "detection elements" (Figure 5).

The results of the FMEA are presented in tabular form. Table VII presents an extract





| Component 2 | Technical function | Failure mode | Possible causes of failure = constraints not complied with | Detection element/ Detection | Possible effects of failure |
|---|---|---|---|---|---|
| Editorial with a word from the mayor | -Inform to raise awareness of risk | The function "Inform to raise awareness of risk" is faulty | The form and content are not complied with | Detection elements based on form: -Relevance of the combination of text and background colors | -The individual does not consult this heading |
| | -Inform to foster the acceptance of risk | -"Inform to foster the acceptance of risk " is faulty | | -Typography of paragraph | -The individual consults this section but does not want to continue reading the DICRIM |
| | | | | - Text Length | |
| | | | | - Structure of the text | |
| | | | | - Presence of photos | |
| | | | | -Type of photos | -The individual does not keep the DICRIM |
| | | | | -Size of the photo | |
| | | | | -Proportion between illustrations, text and whitespace | -The individual does not become aware of the risks present on his |
| | | | | -Typography of Titles | |
| | | | | -Color of titles | |





| Detection elements based on substance : | commune |
| --- | --- |
| -Adaptation of the title to the content of the section | -The individual does not accept the risks present on his commune |
| -Vocabulary | -The resident is surprised by the occurrence of the phenomenon and is in danger |
| -Presence of catch phrases at the top of the page or paragraph | |
| -Synthetic character of paragraphs | |
| -Usefulness of the data presented | |

**Table VII: FMEA table for component 13**



To detect failures of components we identify:  detection elements regarding regulations, form and substance. For form, 13 generic detection elements were identified. They are used for one or other of the 16 DICRIM components according to what they allow to detect. The same approach is followed for the 7 substance detection elements identified.

For the sake of simplicity, only the global causes for failure are given in Table VII (for instance, "the form is not complied with"). Table VIII details these results and exhibits some of these detection elements matched with the constraints they allow evaluating. For instance, all the detection elements in the table must persuade the reader to read the DICRIM by capturing their attention. The presence of photos and diagrams satisfies all the form constraints. It is the only component that expresses the form constraint "Make known that certain actions are specific to a phenomenon". This could be explained by the fact that safety instructions are often represented by pictograms.




| Failure Mode / Detection elements | Persuade reader to read the DICRIM | Capture attention | Promote understanding and recording of information to maintain and fuel risk culture | Encourage understanding and recording of information leading to appropriate action in the face of a major phenomenon | Reassure the reader | Introduce DICRIM function | Explain importance of risk management | Make known that certain actions are specific to certain phenomena |
|---|---|---|---|---|---|---|---|---|
| Typography of paragraph | x | x | x | x | x | | | |
| Relevance of the combination of text and background colors | x | x | x | x | x | | | |
| Text length | x | x | | | x | x | x | |
| Structure of the text | x | x | | | x | x | x | |
| Number of pages of | | | | | | | | |



| | | | | | | | | |
|---|---|---|---|---|---|---|---|---|
| the DICRIM | x | x | x | | | | | |
| Presence of photos/diagrams | x | x | x | x | x | x | x | x |
| Quality of photos and diagrams | x | x | x | | x | | | |
| Map with clear resolution | x | x | x | x | x | | | |
| Map has adapted semiology | x | x | | | | | | |
| The map models the geographical space so that it satisfies the use intended | x | x | | | | | | |
| Map has an orderly and explicit legend | x | x | x | | | | | |

**Table VIII: Detection elements matched with form constraints they allow evaluating.**



The regulation detection elements we listed allow the evaluation of the presence or absence of the required content for each component; in total they are 241. Although this number is high, the production of such detection elements simplifies their appropriation if compared to the tack of localizing them in the National Model which is very long (360 pages) and where requirements about DICRIM are mixed with DDRM regulations and general information about major hazards. Moreover, as they are presence/absence detection elements, the assessment of the whole sets of symptoms can be quite rapid (~ 20 minutes). The completeness of the DICRIM can be verified thanks to them. In our future works, when they will be formalized as indicators and their number will be highly reduced, but they can already be used as they are. Content and form detection elements will also be formalized as indicators and presented in a grid form. An example of those indicators is shown in Table IX. It allows evaluating the length of the component, here the "Editorial" component, regarding number of pages combined with the font size. This is one of the important features to take into account for evaluating its effectiveness.

| Name | IC6 – Component length |
|---|---|
| *Definition* | The aim is to evaluate the length of the whole "Editorial" component |
| *Scale and references* | <br> 10: 1/8 or 1/4 page and the font is of standard size (usually 12) <br> 6: 1/2 page and font size is standard <br> 5: 1/8 or ¼ page and the font is greater than 14 <br> 3: the font is greater than 14 and the length is greater than 1/4 page <br> 2:> 1/2 page or the font is less than 10 |
| *Place characteristic* | At the beginning of the DICRIM - On the page where the editorial is located |

**Table IX : Example of an indicator grid here related to the evaluation of the lengh of the editorial component**

Then all those form and content indicators will be aggregated to form an effectiveness detection model. Several aggregation operators can be used such as weighted average or minimal… In Figure 7, an example of the effectiveness model is showed. It contains indicators with their hypothetical weights, used to demonstrate how a





"bad" score becomes useful to detect where feedback must be applied in order to improve a given components' overall score.

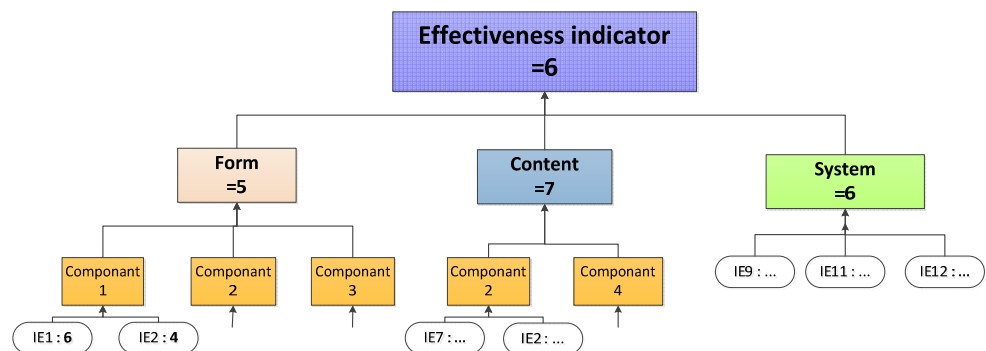

**Figure 7 : Example of the functioning of the future model of effectiveness by indicators aggregation**

An application is leading using regulatory indicators in the next section.

## 5. Application

A database of 30 DICRIMs was built. They were all produced by towns located in the Provence-Alpes-Côte-d'Azur region in France. The DICRIMs were retrieved from municipal websites. A variety of DICRIMs were collected, considering: the number and types of hazards affecting the town, the size of the town in terms of number of inhabitants or surface area, the environment of the town (mountain, sea, countryside) and criteria regarding the document itself were also considered. The DICRIM's were more or less long (from 5 to 56 pages – not linked with the number of risks identified). The average length was 20 pages. Their dates of creation also varied with some more being recent than others (from 2003 to 2015).

In this application, we wanted to analyze compliance with regulations for all the DICRIMs. Thus, we analyzed the presence or absence of all the requirements of the national model provided by the MTES. To this end, we used the regulatory detection elements we listed when implementing the FMEA method (cf. § 4.3). We did not consider mining and volcanic risks because they did not threaten any of the 30 towns studied. That is why 199 detection elements were used during this analysis. Each DICRIM was carefully read and each detection element was filled with the number "1" if the element was present or a "0" if it was absent. The mention "nc" (not concerned) was noted if the hazard asked by certain detection elements was absent in the town studied. The results are presented in table form. An extract from this table is shown in Table X.

Results highlight that there is a big lack of elements in all the DICRIM reports. In fact, the existing elements seldom reach half of the total requirements per DICIRM. For instance, town 1 contains 58 of 128 required elements or town 5 only 40 of 125 required elements ergo in both cases their number is less than half of the elements needed. In conclusion, table IX shows that none of the DICRIMs exposed in the database fulfils the requirements of elements.





Elements relative to hazard safety instructions were those which were most present in the DICRIM database. This observation is interesting because safety instructions are major part of one of the two service functions of a DICRIM. The second most common element concerned "the presentation of the DICRIM's role", an important means of capturing readers' attention.

This analysis also showed that 19 elements required by the national model were not present in any DICRIM in the database, 22 are only present in one DICRIM each time and 18 in two MIMDR. These observations lead to questions regarding the relevance of these elements. In particular, one of them was nearly always missing: that of the role of insurance for each hazard. Some people may think they will be better compensated in case of damages on their property than they will really be. It can unconsciously lead them to less prepare their own safety in prevention when a phenomenon is announced. "History of the risk concerned in the municipality by mentioning the most significant events" was also often absent, which was unfortunate in our own opinion. Mentioning significant events is an effective way of raising public awareness of hazards that threaten the town in which they live. It is also a way of preserving memory of the risk. In addition, the lack of a hazard map and a list of equipment necessary in the case of an event is regrettable. Indeed, to our mind they are also important elements for both service functions of the DICRIM.

The analysis of the presence of elements allows evaluating to what extent the DICRIM does or does not comply with the regulations. However, this does not necessarily provide any conclusion on the effectiveness of these DICRIMs. The use of other non-regulatory, formal and substantive detection elements is necessary to achieve such a measure.

Another question is to understand why some elements were missing and to determine their importance for risk prevention.



| Detection elements/ Towns | Town 1 | Town 2 | Town 3 | Town 4 | Town 5 | Town 6 | Town 7 | Town 8 | TOTAL (in the whole database – 30 DICRIM) |
|---|---|---|---|---|---|---|---|---|---|
| Presentation of the role of DICRIM | 1 | 1 | 0 | 1 | 1 | 1 | 1 | 1 | 27 |
| Major definitions of risk and terms characterizing them | 0 | 0 | 1 | 1 | 1 | 1 | 1 | 0 | 15 |
| Description of flooding event | 1 | 1 | 1 | 1 | 1 | 1 | 1 | 1 | 26 |
| Story of flood risk in the town by | 0 | 0 | 1 | 0 | 1 | 1 | 1 | 0 | 12 |



| Feature | | | | | | | | | Count |
|---|---|---|---|---|---|---|---|---|---|
| mentioning the most significant events | 1 | 1 | 0 | 0 | 1 | 1 | 1 | 1 | |
| Explanations of the flood monitoring implemented | 1 | 1 | 0 | 0 | 1 | 1 | 1 | 1 | 21 |
| Description of existing population flood warning alerts | 1 | 0 | 0 | 0 | 1 | 0 | 0 | 0 | 19 |
| Symbolization of pictograms (floods) | 1 | 1 | 1 | 1 | 1 | 1 | 1 | 1 | 24 |
| Safety instructions relative to flooding and advocated in the national model and in the DDMR | 1 | 1 | 1 | 1 | 1 | 1 | 1 | 1 | 29 |
| ... | ... | ... | ... | ... | ... | ... | ... | ... | ... |



| | | | | | | | | | |
|---|---|---|---|---|---|---|---|---|---|
| Description of the phenomenon "transport of dangerous goods" | 25 | nc | 1 | 1 | 1 | 1 | 1 | 1 | Nc |
| Story of transport of dangerous goods risk in the town by reference to the most significant events | 5 | nc | 0 | 1 | 0 | 0 | 1 | 0 | Nc |
| … | … | … | … | … | … | … | … | … | … |
| Mention of contacts, phone numbers and website links. | 25 | 1 | 1 | 1 | 0 | 1 | 1 | 1 | 1 |





| Mention of each of the minimum items of equipment required: portable radio with extra batteries, flashlight, drinking water, food supply, containment equipment, first aid kit, personal papers, emergency medicine, blankets, extra clothing | 0 | 0 | 0 | 0 | 0 | 0 | 1 | 0 | 17 |
|---|---|---|---|---|---|---|---|---|---|
| TOTAL (/number of elements required) | 58 /128 | 40 /108 | 70 /143 | 82 /144 | 40 /145 | 42 /126 | 66 /106 | 46 /91 | |

**Table X: Application of regulatory detection elements in an DICRIM database (extract)**



## 6. Conclusion and Perspectives

Providing information and communication is essential for raising awareness of risks and disseminating knowledge on their nature and on how to act if a phenomenon occurs. That is why it is crucial to ensure that relevant information and communication is available to those who use them. It allows decisions to be taken without bias. However, few works currently propose to evaluate their effectiveness. Existing works often require a sample population or are expensive to perform. According to the means available, such studies are not always possible. Furthermore, these methods are specific to the sites where they must be applied and are not necessarily generic.  In this study we proposed an analytical methodology that allows identifying communication functions and dysfunctions. It lies at the interface between several existing approaches, from different fields (engineering, advertising, etc.). It was applied with a database of 30 DICRIMs and included regulatory detection elements listed using the FMEA method. These detection elements were then used to analyse existing documents and show their degree of conformity with the regulations. A challenge in the future will be to analyse whether component redundancy is strictly necessary (Table IV). Indeed, we observed that some components fulfilled the same function, raising doubts as to their usefulness. Likewise, the application of the database showed that several elements demanded by the regulations were systematically missing from the DICRIMs studied. This also led to questioning the need for these elements to fulfil the objectives of the document.

The approach is generic and could be applied to other documents, notably for preventive risk management such as the family safety plan intended to help families. Families have to complete it themselves in order to prepare for the possible occurrence of an event. Our approach and its general procedure (EFA and FMEA), service functions, types of constraints, form constraints, and some of regulatory and substances constraints may be applied to the case of family safety plans.

This study is the first step towards a decision support model for the municipality (or specialized consulting firms) in charge of drawing up documents. This model would allow evaluating the effectiveness of existing DICRIMs and identifying the corrective actions needed to improve their effectiveness. In the next step (not presented here), the causes and effects identified with the FMEA method will be used to define models for the quantitative assessment of DICRIM efficiency. Detection elements will become indicators formalized and those will be used as input data in these models. This is an essential step towards the overall goal. Its results are crucial for ensuring the basis of models and for structuring them.





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
