# Peer review of "Analysis of a risk prevention document using dependability"

_Natural Hazards and Earth System Sciences, 2017_

## Referee Comment (RC1) · J. Douvinet (Referee) · 11 Oct 2017

Dear authors

This paper focuses on a specific French document, the DICRIM, and presents an original evaluation on how this can be improved and enhanced for population. In general manner, several sentences should be re-written and authors should be verify the bibliography as several works are note cited at the end of the paper.

In introduction, the 6 French laws are not related to the same objectives and even if the figure 1 is interesting, it could be better to dissociate information / communication

/ preventive ways as well as actors (population, stakeholders...). We do not see also the Knowledge over Existing Data (Portée à Connaissance in French). Data and bibliography should be also added over the idea that "It is also very difficult to establish if the system achieves its purpose in terms of being appropriated by the local population". On the other hand, " Preventive information is also provided through other means (what kinds?) but we chose to focus on DICRIM because it is the main regulatory tool of compulsory form dedicated to the general public that summarizes all risks and their prevention". Authors have to detail the structuration and variery of forms of collected DICRIMs. The aim of this paper is to evaluate the effectiveness of DICRIM, but it should be useful to define what authors assimilate to effectiveness (appropriation by citizens for exemple?)

For the methods to evaluate communication pervasity, one of the major problem is to evaluate the appropriation by citizens of information included in the DICRIM. So connected or ROI methods are useful, surely, but why? Another question is to determine that people regarding the DICRIM will apply satefy guidelines included in DICRIM. How it is possible to measure this ? Several papers have yet shown that konwledge of risk is not interrelated with behaviors, and these reviews should be mentionned?

The content of a DICRIM is yet detailed by other authors. Is it possible to define a graduation for the different document attempted for this DICRIM?

The EFA evaluation is quite shorter. Do the general public is the real destination ? The DICRIM is not only a guide for action and information. It can also be considered as a document to inform over the risk existing in a municipality, over the last CatNat events and over the safety actions. So different informations for different objectives...

The failures model could be summarized in another format (the table is not easy to read)

In conclusion, why this work differ from previous? What king of advices the authors can propose to perform the effectiveness of such document in France?

The component/service actions are qualified by a specific indicator, but a word or a page dedicated to an information conduct to have the same data, so how it is possible to define a good DICRIM (complete and short format : 4p. for example) or a bad DICRIM (strongly detailed, with DICRIM of 74 p. for example in several municipalities...)

For the technical components, only 16 DICRIM have be integrated. Why ? 2 250 DICRIM exist according the BD-DICRIM and more than 5 500 for the French Ministry. So why do not use for example 200 DICRIM?
* * *

---

## Short Comment (SC1) · 21 Oct 2017

This paper aims to evaluate the effectiveness of Dicrim (PACA region, France) by a method based on a dependability approach: External Functional Analysis -EFA. It is a method used in the fields of engineering (industry) and marketing (advertising). In engineering / industry, the method uses models (formulas / equations), algorithms (UML tools) ... to evaluate the failures of the system studied. In this work, the FMEA failures are concretely identified & characterized by a binary (0, 1) criteria (detections elements) approach leading to a global score for each Dicirm. This multi-criteria evaluation is about 6 pages long (p. 26-31) where 4 pages are dedicated to the Table

n°X ! After an introduction of about 3 pages, the state of the art is the subject of 2 pages (p. 5-6) and the detailed presentation of the method used (and applied on a Dicrim as a system) is the object of at least 19 pages (p. 7-25): hence there is a significant imbalance between the two main parts of the article. The main aspect of the work is indicated by the fifth column of Table VII on the page 20. Indeed, the "detection elements" of the "form" of the Dicrim considered are: the relevance of the association of the text with a background, the font size of the text, the presence of photos, the size of the photos, the typography and the color of the titles ... In brief, it is a question of evaluating the visual attractiveness of the document by the specialists (and not by the end users) and not necessarily its understanding (cognitive integration). This attractiveness (visual) relates to known disciplinary fields: semiology & semiotics, study of eye movements, extraction of knowledge (if research on cognitive integration topic) ... with numerous works not taken into account. Why a font size 12 (standard size) and not another value? What values of visual acuity (observer), salience / contrast (observed) ... for a more readable Dicrim? Which rate between text and image to recommend? Recent studies have shown that only a small area (10% or 20%) of a document can be of interest (up to 80%) to the reader compared to the rest of the document (area of interest analysis). Can we summarize the methodological / theoretical part and focus more on the elaboration and evaluation of criteria (with specialists and / or end-user volunteers)? In other words, what is the added value of the methodology section with regard to the application? The methodology part is a big work whose final result (multi-criteria rating) is disappointing. For example, this section could have more relevant results like UML model given the many logigrams and tables developed. Finally on the general form, reading the paper is very difficult because of the interdependencies between the different logigrams and tables. The font size of the text is not equal to 12 (standard size)! The developments of the page 15 ... cannot be justified by only one reference (CEPRI)! Figure 1 is unreadable. This type of figure (logigram) is easy to understand by an "expert" of the French system.

Please also note the supplement to this comment:
https://www.nat-hazards-earth-syst-sci-discuss.net/nhess-2017-311/nhess-2017-311-SC1-supplement.pdf

───────────────────────────────

---

## Author Comment (AC1) · 9 Nov 2017

Authors sincerely thank the referee Mr Johnny Douvinet for his review of our paper and relevant remarks. Please find below a point-by-point reply to your questions. We hope that our responses will agree with your thinks: (Your question is reported with a Q and our response with a R)

Q-In general manner, several sentences should be re-written and authors should be verify the bibliography as several works are note cited at the end of the paper: R-After taking into account your comments, the paper will be corrected again by a native translator. Bibliography will also be verified to eliminate those errors.

[Figure]

Q-In introduction, the 6 French laws are not related to the same objectives: R- Laws written in 1982 that refers to compensating victims of natural disasters, and 1995 (creation of Risk Prevention Plans),will be removed to keep focus on preventive information.

Q-even if the figure 1 is interesting, it could be better to dissociate information / communication / preventive ways as well as actors (population, stakeholders...). We do not see also the Knowledge over Existing Data (Portée à Connaissance in French): R- with the aim of improve the figure 1 we will first differentiate institutional stakeholders and population actors using two symbolizations. Secondly, we will also clearly differentiate documents (DICRIM...) from communications (Public meetings). Thirdly, we will indicate communications made on one hand by mayor and on the other hand, by population using different arrows. Finally, we will add the Knowledge over Existing Data to the figure.

Q-Data and bibliography should be also added over the idea that "It is also very difficult to establish if the system achieves its purpose in terms of being appropriated by the local population" : R-Some references will be added in the introduction to support this affirmation notably regarding behaviours during a phenomenon.

Q-On the other hand, " Preventive information is also provided through other means (what kinds?) but we chose to focus on DICRIM because it is the main regulatory tool of compulsory form dedicated to the general public that summarizes all risks and their prevention" : R-a few description of other means of preventive information as PPR, flood benchmarks will be added in this section.

Q-Authors have to detail the structuration and variery of forms of collected DICRIMs: R-Statistics data about this database will be mentioned earlier, in the introduction, to give more precisions on the structuration of the collected DICRIMs. The aim of this paper is to evaluate the effectiveness of DICRIM, but it should be useful to define what authors assimilate to effectiveness (appropriation by citizens for exemple?) We will give a definition of effectiveness that is clearly missing. Âń Effectiveness is the level

of achievement of planned activities and achievement of expected results (Association Française de Normalisation 2005). It refers to the concepts of function and dysfunction Âż The definition will be added in the paper based on this reference and adapted to DICRIM.

Q-For the methods to evaluate communication pervasity, one of the major problem is to evaluate the appropriation by citizens of information included in the DICRIM. So connected or ROI methods are useful, surely, but why? R-Evaluating the appropriation of information by the population is effectively one of the major challenges of effective communication. Connected methods can be useful in that they will help to determine whether DICRIMs have aroused the interest of people by convincing them, for example, to go to the associated websites (using QR code) for additional information after reading of DICRIM. This element will be added in Section 2.

Q-Another question is to determine that people regarding the DICRIM will apply satefy guidelines included in DICRIM. How it is possible to measure this ? R-Even if "inform on acting appropriately when facing a major phenomenon" is one of the main objectives of a DICRIM, it is true that there is no assurance that people will apply them correctly. First of all because each hazard described in the DICRIM has its own safety instructions which lead to no less than about twenty safety guidelines on average. Some studies are measuring effectiveness of DICRIM by asking people about how they will act in case of a phenomenon but it is impossible to ask a person to memorize twenty different safety guidelines. They should have those guidelines at hand to refer at it when it happens. But in order for the population to appropriate this information, however, it must be ensured that it has reached them, which is not always the case. Surveys carried out on DICRIMs have revealed that among interviewees, some say that they would simply flip through the DICRIM and then throw it away, revealing a flagrant lack of interest. A perspective of our works is to make a survey with two similar sample evaluating an old DICRIM and a new enhanced one to evaluate this information apprehension.

Q-Several papers have yet shown that konwledge of risk is not interrelated with behaviors, and these reviews should be mentionned? : R-It is true that several studies showed that knowledge of risk does not necessarily lead to an appropriate behavior. Sometimes for instance people will take irrational decisions, or decisions based on economic consideration rather than secure (which could be the case when people chose to go to retrieve their car in underground park during floods of Cote d'Azur Region in 2015). Those bias take then precedence on knowledge and information they may have received preventatively. Some bibliographical references (Slovic 1987 ; Khaneman & Tversky 1979 ; Sjoberg 1998. . .) will be added in the paper notably regarding the Figure 6. But other studies showed that information can influence people, like it has been the case for prevention for road safety in France that has consequently increased the number of person wearing seat belt. But this is not the main topic of our research, we did not want to evaluate effectiveness of DICRIM by mesearing if they act correctly thanks to it, we mainly want to make sure that the information is apprehended by the reader. Before the research of "good" actions which is of course the finality, the rise of awareness about major hazards is for instance a crucial question a DICRIM can help to. Some references will be added in our paper to argue your relevant remarks.

Q-The content of a DICRIM is yet detailed by other authors. Is it possible to define a graduation for the different document attempted for this DICRIM? : R-some references discussing recommendations about the content of DICRIM will be added in the paper and compared (Guidebooks by CEPRI, IRMa or articles such as Lowrey W. et al. (2007). "Effective media communication of disasters: Pressing problems and recommendations", BMC Public Health)

Q-The EFA evaluation is quite shorter. Do the general public is the real destination ? The DICRIM is not only a guide for action and information. It can also be considered as a document to inform over the risk existing in a municipality, over the last CatNat events and over the safety actions. So different informations for different objectives... : R- EFA has made it possible to obtain 2 main service functions which are identified with a high granularity (macro vision) and which allow the different objectives of the DICRIM

to be synthesized. These are then broken down into each of the technical functions associated with each component (Table 4). When we speak of "information" (FS1) we include CatNat events, for example, which are just one of the means to inform people (if an event has occurred in the past, it will happen again) and who will participate in maintaining the memory of risks. DICRIM is a central information document on the risks affecting the municipality from which it is derived and which is addressed to all.

Q-The failures model could be summarized in another format (the table is not easy to read) : R-Instead of showing failures model with a table we will use a graphic representation in the form of a bowtie schedule. It will represent a scenario with mode of failures in the centre, its causes at its left and effects at its right both linked with arrows. Technical functions will be removed as they are already showed earlier. Detection elements will be attached to mode of failures with another symbolization. We will also add a description to give an example and facilitate its reading. For instance: the component "Editorial with a word from the mayor" owns the technical functions "inform to raise awareness of risk" and "inform to foster the acceptance of risk". If one of those functions is faulty, it is because the form and content are not complied with. To detect this failure some elements are helpful based on form and content (typography, type of photos, usefulness of the data presented. . .). Detect those failures are crucial because they can lead to possible effects. For instance, the individual consult this section but does not want to continue reading the DICRIM and then it is possible that he does not become aware of the risks present on the commune he lived and that he is surprised by the occurrence of a phenomenon and in danger.

Q-In conclusion, why this work differ from previous? What king of advices the authors can propose to perform the effectiveness of such document in France? : R-Previous work evaluating the effectiveness of a DICRIM is based on ad hoc surveys carried out at the level of given municipalities or territories, on a given document. They also need significant human and material resources. This work differs from previous because it allows to obtain a generic approach based on systemic and analytical reasoning applied

to all DICRIMs (see Figure 3). The purpose of the article is to explain this approach by stressing the analysis phase which then structures the models. The example of compliance with the law is given to illustrate. We will insist more on these aspects of originality and the contribution of this work in the article.

Q-The component/service actions are qualified by a specific indicator, but a word or a page dedicated to an information conduct to have the same data, so how it is possible to define a good DICRIM (complete and short format : 4p. for example) or a bad DICRIM (strongly detailed, with DICRIM of 74 p. for example in several municipalities...) : R-A same data showed by a word or a page will not have the same impact on the reader. The format is often the information gateway. If the reader is not really interested in the subject, it is possible that he/she may be discouraged from reading an entire page where just a few synthetic sentences would have been sufficient. Good and bad DICRIMs are defined in indicators that were made based on communication expert recommendations (regarding the form). High notes on the scale correspond to good characteristics and low scores to bad ones that must be improved.

Q-For the technical components, only 16 DICRIM have be integrated. Why ? 2 250 DICRIM exist according the BD-DICRIM and more than 5 500 for the French Ministry. So why do not use for example 200 DICRIM? R-We made and used a database of 30 various DICRIM coming from Paca Region. For each of them, the 16 components that composed a DICRIM identified thanks to the EFA method and based on sections required by the law (Police and protection actions; General safety instructions; Mapping 1/25.000th ; Emergency phone numbers…) were analysed. We considered that this number was enough to test the feasibility of the method; it was not used for statistical analysis purposes where a higher number of DICRIM would have been required. It is true that it will be interesting in the future to test the methodology on more DICRIM to improve it.
* * *
2017-311, 2017.

---

## Author Comment (AC2) · 9 Nov 2017

We sincerely thank the referee Mr K Serrhini for his review of our paper and his relevant remarks. Please find below a point-by-point reply to your questions. We hope that our responses will agree with your thinks: (Your question is reported with a Q and our response with a R)

Q- In this work, the FMEA failures are concretely identified & characterized by a binary (0, 1) criteria (detections elements) approach leading to a global score for each Dicirm.

R- Only regulatory compliance detection elements are characterized by a binary criteria

approach: compliance is relative to the presence/absence of the components required by the National Model of the Environmental Code. Form and content detection elements will be characterized by a notation scale (0-10 such as shown in Table x) and all those notes will finally be aggregated leading to a global score for the document. A sentence will be added to clearly indicate these two kinds of approach

Q- This multi-criteria evaluation is about 6 pages long (p. 26-31) where 4 pages are dedicated to the Table nâŮęX !

R- Table n°X will be adjusted to reduce the number of pages dedicated to and to facilitate its reading.

Q- After an introduction of about 3 pages, the state of the art is the subject of 2 pages (p. 5-6) and the detailed presentation of the method used (and applied on a Dicrim as a system) is the object of at least 19 pages (p. 7-25):

R- hence there is a significant imbalance between the two main parts of the article We will rebalance the two parts by reducing the introduction and increasing the state of art. The presentation of the method will also be changed as it is explained in next answers (please see notably Response 5).

Q- The main aspect of the work is indicated by the fifth column of Table VII on the page 20. Indeed, the "detection elements" of the "form" of the Dicrim considered are: The relevance of the association of the text with a background, the font size of the text, the presence of photos, the size of the photos, the typography and the color of the titles ... In brief, it is a question of evaluating the visual attractiveness of the document by the specialists (and not by the end users) and not necessarily its understanding (cognitive integration).

R- FMEA allows identifying two types of detection elements: form and substance. That is why it was as much about evaluate the visual attractiveness of the document as it was about its understanding (evaluated by substance elements such as "useful data"

for instance).

Q- This attractiveness (visual) relates to known disciplinary fields: semiology & semiotics, study of eye movements, extraction of knowledge (if research on cognitive integration topic) ... with numerous works not taken into account. Why a font size 12 (standard size) and not another value? What values of visual acuity (observer), salience / contrast (observed) ... for a more readable Dicrim? Which rate between text and image to recommend? Recent studies have shown that only a small area (10% or 20%) of a document can be of interest (up to 80%) to the reader compared to the rest of the document (area of interest analysis).

R- Some references issued from those disciplinary fields (Backer, Bost, Cobb 1996, Chesneau 2006, Roulet 2008,...) will be added in the text to justify the list of detection elements identified. In the same vein, a text (supported by references) will be added to support the scale references given in Table IX (font size 12) and others you propose such as "rate between text and image" or "values of visual acuity (observer)"

Q- Can we summarize the methodological / theoretical part and focus more on the elaboration and evaluation of criteria (with specialists and / or end-user volunteers)? In other words, what is the added value of the methodology section with regard to the application?

R- Our two main results are: -the development of a method of analysis leading to various types of criteria (compliance, form, substance) which is generic and applicable to other documents of prevention. -the application of the method to the DICRIM with an example of the use of the compliance detection elements To better exhibit the added value of our works, we will change the plan of the article. We will remove the DICRIM from the presentation of the methodology because our methodology is a systematic reasoned and reusable approach, the functions remaining the same for other prevention documents. DICRIM will be presented after as an application. The Figure 3 will be located at the beginning of the section of the global approach and the section "system

studied" will be showed after it. We will also add a hypothesis that the methods of dependability analysis can be applied to the case of the effectiveness of risk prevention documents the paper is answering. In this way, those changes will rebalance the number of pages of the different sections (complement to Response 3).

Q- The methodology part is a big work whose final result (multi-criteria rating) is disappointing. For example, this section could have more relevant results like UML model given the many logigrams and tables developed. Finally on the general form, reading the paper is very difficult because of the interdependencies between the different logigrams and tables. The font size of the text is not equal to 12 (standard size)!

R- Some tables are not essential in the paper and will be removed to simplify the reading (table 5 and maybe 7). Figure 5 will be improved as an UML model given the link between tables. As indicated in our reply to the first referee M Douvinet based on its remark, the FMEA table will be removed and replaced by a graphic representation in the form of a bowtie schedule showing a set of scenarios .

Q- The developments of the page 15 ... cannot be justified by only one reference (CEPRI)!

R- Some references will be add to justify those developments as (Slovic 1987 ; Khaneman & Tversky 1979 ; Sjoberg 1998. . .) regarding cognitive bias.

Q- Figure 1 is unreadable. This type of figure (logigram) is easy to understand by an "expert" of the French system

R- We will apply some advices suggested by M. Douvinet to improve this figure. We hope that it will be easier to understand even by a non-expert of the French system because it is an important figure to set the context, to locate the DICRIM in this prevention management and also to show its complexity.

---

## Short Comment (SC3) · 17 Nov 2017

This is a multi-author comment by: A Almerini, A Angle, A Bae, N Belew, J Casselman, S Devendran, D Dusseau, A Evengaard, D Farone, M Feng, X Fonseca-Morales, T G Hamm, R Heath, A Ho, Y Ho, L Hoffman-Hernandez, E Jeong, S Joshi, C Lang, A Liu, B Llamanzares, D Ng, M Nielsen, I Nomura, L Pawar, J Payne, R Cohen, M Ruid, A Schimel, S Schwager, A Soriano Quevedo, N Turner, M Vignes, Y M Xu, Y Zhang

General comments:

Overall, this paper provides a solid analysis of the shortcomings of DICRIM and 3

modes of analysis that could improve the risk prevention document and its use by mayors in France. The author gives a good explanation of sources that have already explored this topic. It would be of interest to the reader to present a broader background explaining why the DICRIM are not efficient at the moment.

Some graphics were visually pleasing, and easily understandable. Others had too many components and colors to the point that a quick meaning cannot be drawn from the image. We suggest to select the key points and components that cannot be explained better in text, and simplify the graphics to emphasize these components. Many of the figures don't have captions and are difficult to understand, and the formatting of many of the tables make the presentation of information difficult to understand.

The methods section should have more detail on the process of the research. For instance, by saying that the DICRIM was "carefully read" does not entirely explain how many times it was read, who read them, was it the same person each time, or how to identify the different factors in the DICRIM. If different people read different DICRIMs, different interpretations would have an impact on the outcomes and conclusions.

We suggest to connect each measurement criteria to the cognitive biases and heuristics of the town, instead of addressing them in one paragraph in the paper. One of the reasons used to justify why the DICRIM needed to be evaluated was to encourage better risk prevention by the French public, but the authors have not addressed research that suggests better information does not necessarily promote increased action. In the conclusion, they state supplying necessary information allows information to be taken without bias. This seems to overly simplify. Making decisions without bias is virtually impossible.

The article could benefit from more analysis of the results instead of just a presentation of results. Also, the results are provided in a chronological experimental context as opposed to a logical sequence. It would be very helpful if the useful elements which were missing from the DICRIM were presented more prominently as the findings of the

article.

Lastly, the target audience of this article is not clear. It seems unlikely that policy makers would be able to take valuable information from the analysis, as gleaning the most important take-away messages is difficult. It would be helpful to incorporate more discussion on how this could be used by local policy makers; we suggest soliciting feedback from a test group of such stakeholders to hear their comments. What is the definition of effectiveness? Improving effectiveness of the DICRIM is one of the main points of article, so defining success more clearly would be helpful. It is also a little unclear how this approach would have equal relevance outside of France.

While we have found the content of the paper innovative and relevant, we have had reservations about its style. The paper suffers from heavy use of jargon, and a lot of repetition. Many of the sentences feel like "stops/starts" and disjointed (e.g. 5. Application, p. 26), rendering it difficult for the reader to comprehend the authors' points. Oftentimes, the authors utilize pronouns as subjects and we were confused as to what "it"s and "they"s specifically referred to. If the authors use a diversity of sentence structures, this can make their content a lot more cohesive. In general, shortening and condensing the article would be helpful, to allow for an easier, more concise read.

Specific Comments:

We don't feel that tables should be supplementary to information provided; they should stand alone and be easily interpreted by the reader. If they are supplementary, they can be added in the appendix. Table IV is not a standalone graphic and needs further information in order to be understood. More information needed for page 27's final analysis.

Tables in the paper are poorly stylized both in terms of graphics and in terms of the text. e.g. Table III on page 14 attempts at outlining technical functions of services. Two of said functions are "Inform to raise awareness of risk (TF1)" and "Inform to fuel knowledge on risk (TF2)". It is unclear what "fueling knowledge" means. And for a

reader who does not look for an explanation in the article, it is not clear how TF1 and TF2 are different.

On p. 11, Table I, what are the ellipses (...) at the very bottom? Is part of the table missing? What does the "(extract)" mean in the end of the figure legend? There are also some spacing issues with text in figures and tables (e.g. p. 11, Table I).

Table IV: Need to rewrite the technical functions and components.

P. 18 Table V: very difficult to read the text and format of information. For example, instead of "x"s denoted satisfaction of a component, authors could fill in the entire box with black for a stronger visual and easier understanding.

On p. 3, Fig. 1, it is difficult for a person who is red-green color blind to read the boxes denoted as "urban planning" components; easy change. Otherwise, an infographic may be a suitable substitution.

Figure 3: Not entirely clear what has been done previously vs. what will be done in this paper.

Bullet points on bottom of page 5 could be converted to paragraph form. Some of the figures, e.g. figure 2 have features, e.g. dotted lines, which are assumed to convey certain information but the meaning is not explained.

In page 21 the cells are blank. In 4.2.4- the title Constraints/Components is unclear. There were some parts that made claims without providing supportive statements, e.g., line 24, pg 4.

Consider citing official disaster statistics rather than secondary sources (Introduction sites Huffingtonpost & BBCNews).

In terms of style, we suggest to avoid 'justified' text in charts especially, as it creates strange spacing of words that are difficult to read.

Technical corrections:

Figure 1 needs a description added.

3.2 line 35 does not specify the experts involved and "classical specialist" needs to be defined.

Figure 2 has "literature" spelled incorrectly.

4.2 line 10: TFA is not a common term and needs to be defined.

4.1.2 doesn't have complete sentences

Some of the acronyms are difficult to remember and lengthy; an Appendix at the beginning or end with a list of all acronyms to refer to would be helpful.

Consistent font size would also be helpful for the reader's flow.

On p. 4, line 30, we cannot understand whether you are telling the reader to communicate the message clearly, or that they are communicating their own message clearly? Please clarify.

Page 2: "This is highly significant as human behavior during major disasters is influenced by their own knowledge of risk 15 (Enrico L. Quarantelli 2008). However, it equally depends on their cognitive bias (overconfidence, risk control delusion, denial, irrationality, etc.) (Kahneman and Tversky 1979) on the particular situation they face and more precisely on how they perceive that situation (Matt Dombroski 2006). " The wording in this excerpt is confusing. We do not know what "it" means in "it equally depends". It is not helpful to the reader to have a list of cognitive biases in parentheses; we suggest to break that apart and go into depth.

Page 3: "In 2004, the update and modernization of the 1987 law takes place. " - clunky wording.

P2 L23 "provide" -> "providing"

P3 L10 "al" -> "and"

P4 L23-24 "they are not thinking to communicate to the Great Public" -> "they are not designed for communication with the greater public"

When referencing material there are times when instead of et al, the authors wrote 'and al' Pg 11, line 5- seems unnecessary.

In abstract, line 8- "Given than" should be "given that"

Table on page 31 is missing text in multiple boxes.

Tables on pages 28-31 could be better designed. The spacing could be tighter to make the tables fit in the horizontal direction, and color could help improve the readability

Spelling error in the title: depend ability should be "dependability". Could effectiveness and dependability be defined in the abstract or early in the introduction?

Table VII: what does the word "faulty" mean, what are the criteria to say that we are effectively in a failure mode p15 L11-12 inversion of TF3 and TF4 in regard to Table III

We suggest to avoid the frequent use of parentheses, as this helps with the flow of ideas to the reader. Line 2, page 2, for example, "year" needs to be plural.

Introduction, page 3 line 10 "behavior al" should be "behavioral". Also the use of the word "behaviour" was used in conjunction with behavior. Good to keep consistent.

Page 8, line 10 "evaluation is conduced" should be "conducted"

Figure 2, "litterature" should be "literature"

Page 22 "For the sake of simplicity", perhaps the simplicity is too simple? Making it challenging to understand.

Page 17, line 9 "fulfil" should be "fulfill"
* * *

---

## Author Response (AR1)

**Point-by-point reply to the comments**

Our manuscript has been modified following the different requests and suggestions of the referees. A point-by-point reply for each referee were uploaded in November. Remarks from the Short Comment of E. Coughlan de Perez and other researchers were also considered in this new version of manuscript. This point-by-point reply explain changes made relative to requests asked by the editor this way:

(1) a better clarification (for international readers) what the analysed French system is about.
    *-Figure 1 has been modified into an infographic to ease its understanding. We also added some sentences in the introduction and the state of the art to better explain French system notably regarding risk preventive information made for General Public.*

(2) it would be good to shorten the material a bit so that the structure will become more clear.
    *-The entire plan of the article was revised and modified, and the different parts rebalanced, in order the structure and the content to become clearer for readers.*

(3) several issues can be improved so that the entire work becomes more accessible.
    *We made some changes based on referee's reviews and considerations of the Short Comment: Tables and Figures were improved, references about different topics were added, some details in different parts of the article were also added to precise them, the problematic, the methodology and the application to the DICRIM were better explained and some technical changes were also made.*

We upload a marked-up manuscript version showing the changes regarding those three remarks. We sincerely thank the editor for these recommendations and hope that this new version meets its expectations.

Best regards,

L. Ferrer, C. Curt, JM. Tacnet.

[revised manuscript text omitted]

**Commenté [LF2]:** -Some sentences have been added relative to the French system of risk prevention

-Problematic of the paper has been reformulated to better explain the objectives of the research

-Definition of the term "effectiveness) has been added

[Figure]

Commenté [LF3]: The figure has been redesigned in the form of an infographic to improve its understanding

[Figure]

**Figure 1[1]: French regulatory prevention information throughout the town: Territorial Coherence Schemes (SCoT); Knowledge over Existing Data (PAC); Flood Risk Prevention Plan (PPRI); Development and Planning Guidelines (OAP); Urban Local Plan (PLU); Departmental Document on Major Risks (DDRM); Municipal Information Document on Major Risks (DICRIM); Public notices; Purchaser Tenant Information (IAL); Familial Plan for Safety Layout (PFMS); Public meetings; Flood Marks**

This is highly significant as human behaviour during major disasters is influenced by their own knowledge of risk (Quarantelli 2008). Behaviours, however, are not necessarily the result of logical and rational reasoning. Attitudes of denial or risk underestimation, often found among inhabitants exposed to natural hazards, are commonly interpreted as an "addiction to risk", and sometimes judged as an incomprehensible unawareness in the face of a known and visible danger. This is the cognitive dissonance phenomenon described by (Festinger 1957). For example, a person living in an area that they know to be exposed to a natural hazard is clearly in cognitive dissonance (Schoeneich and Busset-Henchoz 1998). This makes it possible to better understand certain attitudes of scepticism, or even mistrust, towards technical or scientific studies and information. Indeed, to reduce the psychological discomfort caused by dissonance, individuals can act on their risk representations.
* * *
[1] Prefect is a government representative of an area or department (France territorial division). It is thus responsible for public order, ensures the application of laws and regulations and verifies that the local authorities (Town Hall or EPCI which is a grouping of town halls) respect them as well.

Cognitive biases can then substitute knowledge of individual risk as an act of denial, which consists in trivialising the risk involved and is similar to a phenomenon of disbelief (Weiss, Colbeau-Justin, and Marchand 2006) or an illusion of control, which consists in believing that one has control or influence over external or random events (Langer 1975). (Kahneman and Tversky 1979) and then (Slovic, Fischhoff, and Lichtenstein 1982) argue that people form their judgments heuristically, based on their experience, habits, or cultural traditions that enable them to construct their cognitive risk representation. In fact, people fear that they will find themselves unsecured and that their certainties will be undermined by consulting information on the risks affecting their living environment.

Residents are also generally thought to have a lower human vulnerability than tourists, as risk awareness is often attributed to locally rooted populations (Hubert and De Vanssay 2005). However, in the sample of victims of floods in the Mediterranean between 1998 and 2011 studied by (Boissier 2013), only 30 out of 203 people were not residents. Even though several other factors may have to be considered, non-residents are less vulnerable because they are more prone to taking instructions, more respectful of evacuation orders and less inclined to take risks (going to school to pick up their children, saving personal belongings because less property needs to be saved, etc.) as pointed out by (Ruin 2007).

Commenté [LF4]: Section added containing bibliography based on recommendations of referees (notably the fact that knowledge of risk is not necessarily interrelated with hehaviours)

[revised manuscript text omitted]
" is one of the main objectives of a DICRIM, it is true that there is no assurance that people will apply them correctly. First of all because each hazard described in the DICRIM has its own safety instructions which lead to no less than about twenty safety guidelines on average. Some studies are measuring effectiveness of DICRIM by asking people about how they will act in case of a phenomenon but it is impossible to ask a person to memorize twenty different safety guidelines. They should have those guidelines at hand to refer at it when it happens.

But in order for the population to appropriate this information, however, it must be ensured that it has reached them, which is not always the case. Surveys carried out on DICRIMs have revealed that among interviewees, some say that they would simply flip through the DICRIM and then throw it away, revealing a flagrant lack of interest.

Commenté [LF7]: A paragraph has been added to show some limits of the DICRIM

[revised manuscript text omitted]

Commenté [LF13]: FMEA is presented thanks to a bow tie form instead of a table to improve its understanding

An editorial with a word from the mayor is defective when its functions "Inform to raise awareness of risk" and "inform to foster the acceptance of risk" are faulty. It occurs when form and content of the editorial are not complied with constraints. To detect those failures form detection elements and content detection elements are used such as evaluate the typography, the text length, the vocabulary... Those failures can have some negative effects. For instance, as the editorial is not effective, the reader does not consult this section, it may not want to continue to read the DICRIM or even not accept the risks present on its town.

Commenté [LF14]: A description of the figure 6 has been added

To detect failures of components we identify: detection elements regarding form and content. The scales of indicators are based on literature and experts. These scales require the mobilization of different disciplines such as in semiotics with for example the improvement of colours (Chesneau 2006), or the study of the zones of interest of an image. (Judd, Durand, and Torralba 2012) study, for example, the ability of visual salience models to predict areas of interest in an image. (Fabrikant, Hespanha, and Hegarty 2010) study how visual salience and training to cartographic reading impact the effectiveness of a map. Using an experimental protocol studying eye movements, feeling and answering volunteers' questions on different types of maps, (Palka 2015), for example, showed that the most useful maps are characterized by a quantity of information contained in them. The location of the fixations has also allowed him to define which areas of the map attract the eye with their graphic or informational characteristics. This knowledge from different fields will in fact be exploited in more detail in our future work during the development of the indicators.

Commenté [LF15]: some references have been added to give some indications on the construction of indicator milestones

[revised manuscript text omitted]

---

## Author Response (AR2)

**Author's Response**

Dear Sir or Madam,

We are very pleased to learn this decision and we sincerely thank the editor for having make it. We also thank the referee 1 for its positive point of view about our works. Regarding their suggestions, we have added a table of all abbreviations used at the end of the article.

Best regards,

L. Ferrer, C. Curt, JM. Tacnet.